# Expository Text Generation: Imitate, Retrieve, Paraphrase

**Nishant Balepur**    **Jie Huang**    **Kevin Chen-Chuan Chang**

University of Illinois at Urbana-Champaign, USA

{balepur2, jeffhj, kcchang}@illinois.edu

## Abstract

Expository documents are vital resources for conveying complex information to readers. Despite their usefulness, writing expository text by hand is a challenging process that requires careful content planning, obtaining facts from multiple sources, and the ability to clearly synthesize these facts. To ease these burdens, we propose the task of expository text generation, which seeks to automatically generate an accurate and stylistically consistent expository text for a topic by intelligently searching a knowledge source. We solve our task by developing IRP, a framework that overcomes the limitations of retrieval-augmented models and iteratively performs content planning, fact retrieval, and rephrasing. Through experiments on three diverse, newly-collected datasets, we show that IRP produces factual and organized expository texts that accurately inform readers.[1]

## 1  Introduction

Expository writing intends to inform a reader about a topic in a clear and logical manner (Weaver and Kintsch, 1991). Such text is highly prevalent online, appearing in various forms such as university descriptions, medical information, and Wikipedia articles. Despite its importance, writing expository text is a difficult task, requiring the author to carefully plan content for the text, obtain facts from multiple sources, and rephrase the facts so the text flows smoothly (Thomas et al., 1987; Davis and Winek, 1989; Santos and Semana, 2015). Although it requires effort, expository writing is vital for making complex information accessible to readers.

To ease these burdens, we propose the task of *expository text generation*. As seen in Figure 1, the task uses an input title or topic (e.g. college name) and a knowledge source of topic-related factual sentences to create a multi-sentence expository text

Figure 1: Expository texts as college descriptions produced by IRP, 5-shot LLaMA equipped with DPR, and RAG, compared to the ground truth. All models use the topic and corpus as inputs. Differently highlighted text indicates significant errors. Bold indicates similar style.

output (e.g. college description). The goal of expository text generation is to provide readers with *accurate* and *organized* information for the topic.

To facilitate the goals of accuracy and organization, we require the generated output to contain **up-to-date facts** found in the knowledge source, and maintain a **consistent style** dictated by the expository document domain. For example in Figure 1, the gold college texts are organized and phrased similarly, discussing the institutions' founding, enrollment, setting, and size. However, finding these facts is nontrivial, as they may be scattered in the knowledge source. Further, sentences with the required style may not exist in the knowledge source, so the style must be learned from expository texts in the training set. Thus, expository text generation models must tackle both challenges of (1) obtaining the dispersed, relevant facts from the knowledge

---

[1]Code is available at https://github.com/nbalepur/expository-text-generation.

source; and (2) faithfully rewording said facts in a learned style of the expository document domain.

Large language models (LLMs) (Brown et al., 2020; Touvron et al., 2023) cannot directly be applied to our task, given their inability to search for up-to-date information in corpora. Hence, a better solution is to leverage retrieval-augmented LMs (Lewis et al., 2020b; Izacard et al., 2022). However, these models tend to produce inaccurate expository texts. For example, in Figure 1, we find that RAG and LLaMA with DPR generate texts that resemble the style of college descriptions, but hallucinate several facts, such as the institution's founding and enrollment, weakening the credibility of the text.

These issues can be ascribed to two limitations of retrieval-augmented LMs. **First**, these models create text all at once rather than sentence-by-sentence, risking factual errors due to the challenge of modeling long-range dependencies (Maynez et al., 2020a; Ji et al., 2022). **Second**, these models may fail to find nuanced facts in the knowledge source, since they use the generic input topic as the query (Lewis et al., 2020b). For example, querying with a university name will likely not result in the retrieval of a nuanced fact like the university's ranking. However, if we perform sentence-level content planning (Hua and Wang, 2019) (e.g. plan that the next sentence should look something like: "*The University of Illinois is ranked #32*"), we can create fine-grained queries. Although these fine-grained queries may contain hallucinated facts (e.g. an incorrect university ranking), they will also include high-quality information-seeking keywords (e.g. "is ranked") that increase the chance of retrieving the correct facts, reducing factual errors.

To overcome these problems, we design a framework named **Imitate, Retrieve, Paraphrase (IRP)**. Rather than generating text all at once, IRP operates at the sentence level and iteratively uses three key modules: the Imitator, Retriever, and Paraphraser. First, conditioned on the text that has been already generated for the output (or initially, a user-given prefix), the **Imitator** produces a stylistic content plan that outlines the next sentence of the output. The stylistic content plan may contain hallucinated entities, but it will correctly outline *which entities* should be discussed, making it an effective query for retrieving facts to include in the next sentence. To create this plan, the Imitator is trained to mimic the style of expository texts in the training set.

Next, the **Retriever** uses the content plan to obtain said facts from the knowledge source. Finally, to generate the next sentence, the **Paraphraser** synthesizes the retrieved facts in the style of the content plan. This sentence is appended to the output document and the process is repeated until the Imitator decides that the output is complete. Overall, IRP generates text sentence-by-sentence and retrieves facts with fine-grained queries to preserve factuality, while the Paraphraser maintains the style of the expository document domain. This design addresses the issues of retrieval-augmented LMs, resulting in factual expository texts that adhere to the style of the domain (shown in Figure 1).

We study the effectiveness of IRP on three diverse, newly-collected datasets: university descriptions, medical drug information, and computer science history Wikipedia articles. Through extensive experiments, we show that IRP produces more factual expository texts compared to existing models. Further, human judges find the outputs of IRP to have the best balance between style and factuality. Our contributions can be summarized as follows:
**1)** We introduce expository text generation, which aims to generate a factual and organized document that clearly informs readers about a specified topic.
**2)** We develop the **IRP** framework to address the challenges of expository text generation. IRP iteratively and explicitly performs the key steps of content planning, retrieval, and paraphrasing.
**3)** We curate three new, diverse datasets to facilitate research on factuality and style in text generation.
**4)** In our experiments, we show that IRP produces highly factual texts that adhere to the style of the expository document domain. Further, we conduct an ablation study and an error analysis to suggest future paths for expository text generation research.

## 2 Related Work

### 2.1 Retrieval-Augmented Generation

Retrieval-augmented generation (RAG) models combine the strengths of retrievers and generators, and have been used for several knowledge-intensive tasks (Petroni et al., 2021; Lewis et al., 2020b; Mao et al., 2021; Wang et al., 2021). Recent RAG models aim to improve the retriever with re-ranking (Cao et al., 2018a; Ren et al., 2021; Fajcik et al., 2021) and the generator by incorporating the evidentiality of passages (Asai et al., 2021).

Expository text generation is a knowledge intensive task, as it leverages a knowledge source. How-

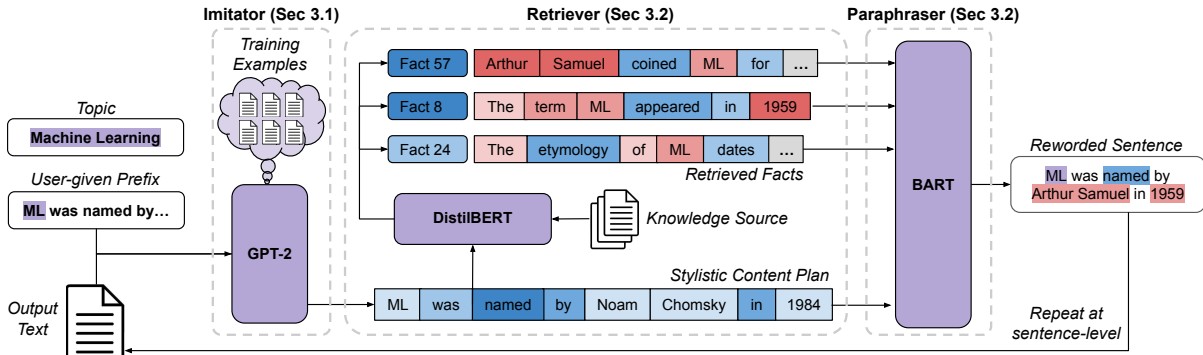

Figure 2: Overview of IRP. First, the Imitator (GPT-2) creates a stylistic content plan that outlines the facts to include in the next sentence. Next, the Retriever (DistilBERT) uses the content plan to find the outlined facts from the corpus. Finally, the Paraphraser (BART) rewords these facts in the style of the content plan. This sentence is appended to the output, which is used as the next prefix for the Imitator. These steps are repeated sentence-by-sentence.

ever, the IRP model has two differences from RAG models. First, to retrieve facts for expository text generation, IRP uses learned, fine-grained queries in the form of stylistic content plans, while typical RAG models use the document title as a query. Second, IRP is an iterative RAG model, meaning that it generates text sentence-by-sentence, and attends to shorter pieces of text at a time. Through ablation studies, we find that both of these design choices improve the performance of IRP (§5.3).

Very recent and contemporaneous RAG models have been designed that iteratively retrieve facts, similar to IRP (Trivedi et al., 2022; Jiang et al., 2023). However, IRP uses smaller LMs trained for expository text generation, while these techniques are prompting methods for large LMs. To be used for our task, these methods require extra engineering to synthesize facts from multiple sentences in a consistent style, so we do not compare with them.

## 2.2 Factuality and Style in Text Generation

Factuality and style are two important areas of text generation research. Notable methods to improve factuality incorporate token constraints and re-ranking (Mao et al., 2020; Zhao et al., 2020; King et al., 2022), modified training data (Cao et al., 2018b; Matsumaru et al., 2020), and custom training objectives (Cao and Wang, 2021; Dong et al., 2022). Recently, researchers have enhanced factuality by post-editing generated text (Dong et al., 2020; Cao et al., 2020; Balachandran et al., 2022).

To control style in text generation, previous works have leveraged stylistic exemplars (Cao et al., 2018a; Wei et al., 2020; Wang et al., 2022). More similar to IRP are models that closely adhere to the stylistic guidance, which have been explored

in controllable paraphrase generation (Chen et al., 2019), data-to-text generation (Lin et al., 2020), and keys-to-text generation (Brahman et al., 2022).

Overall, expository text generation combines the challenges of (1) searching a knowledge source, (2) preserving factuality, and (3) maintaining a consistent style into a single knowledge-intensive task.

## 2.3 Summarization

Although summarization and expository text generation both generate short texts, these tasks have inherently different objectives. Summarization focuses on *condensing information* and thus, the output is a distilled version of the input (Nenkova and McKeown, 2011). In expository text generation, we seek to *synthesize facts in a consistent style*, where the style is found in examples of expository texts, and the facts are dispersed in the knowledge source. Hence, expository text generation aims to synthesize **new** content with a focus on factuality and stylistic consistency, while summarization creates condensed forms of **existing** content.

## 2.4 Wikipedia Generation

Another task that shares similarities with expository text generation is Wikipedia generation (Sauper and Barzilay, 2009), which seeks to automatically create Wikipedia articles from article titles. Typically, this is achieved by retrieving input text from the web followed by re-ranking and generation steps (Banerjee and Mitra, 2015, 2016; Liu et al., 2018; Pochampally et al., 2021). However, these models are often tailored for specific Wikipedia domains. Expository text generation is a much broader task, encompassing more domains than just Wikipedia articles, with the unifying character-

istics of factuality and stylistic consistency.

## 3 Method

The inputs for expository text generation include 1) a topic $t$ for the expository text and 2) a corpus of factual sentences $\mathcal{C} = \{x_j\}$ related to topic $t$. We fix the corpus $\mathcal{C}$ to compare models, but in practice, $\mathcal{C}$ can be acquired in real time for up-to-date facts. To guide the initial generation, IRP also uses 3) a sequence of words $r = \{r_k\}$ to prefix the expository text. Using these inputs, IRP aims to produce a sequence of sentences $\mathcal{D} = \{y_i\}$ to comprise the expository text. The text $\mathcal{D}$ must contain accurate facts about $t$ from $\mathcal{C}$, and must be presented in a consistent style dictated by expository texts in the training set $\mathcal{T} = \{\mathcal{D}_1, ..., \mathcal{D}_n\}$.

As illustrated in Figure 2, IRP leverages three components: 1) a style Imitator $p(y_i|y_{1:i-1})$ that generates a stylistic content plan $y_i$ for the next sentence in the expository document, based on the current state of the output $y_{1:i-1}$ (or prefix $r$ in the first iteration); 2) a Retriever $p(x_j|y_i)$ that returns the top-$k$ factual sentences $x \subseteq \mathcal{C}$ most related to the content plan $y_i$; and 3) a Paraphraser $p(z|x, y_i)$ that combines the semantics of $x$ and the syntax of $y_i$ into a reworded sentence $z$. We will describe each of these modules, followed by how they are combined and trained for the full IRP model.

### 3.1 Imitator

To find facts for the next sentence of the expository text, we must first create a query to retrieve such facts. Hence, the Imitator $p(y_i|y_{1:i-1})$ generates a content plan $y_i$ in the style of the expository document domain for the next sentence in the output, conditioned on the current sentences in the output $y_{1:i-1}$ (or the prefix $r$ in the first iteration). To do so, we seek to imitate the expert content planning of expository texts in the training set, achieved by minimizing the cross-entropy loss of token prediction (i.e. language modeling loss) for the expository texts in the training set $\mathcal{T} = \{\mathcal{D}_1, ..., \mathcal{D}_n\}$, i.e., $\forall w_j \in \mathcal{D}_d, \forall \mathcal{D}_d \in \mathcal{T}$:

$$\lambda_{imit} = -\sum_{d=1}^{n} \sum_{j=1}^{|\mathcal{D}_d|} \log p(w_j|w_1, ..., w_{j-1}). \quad (1)$$

We leverage GPT-2 (Radford et al., 2019) to minimize $\lambda_{imit}$ through causal language modeling.

During each iteration of IRP, we create the stylistic content plan $y_i$ from sentences $y_{1:i-1}$ (or prefix $r$), by first flattening $y_{1:i-1}$ (or $r$) into a list of tokens $s = [s_1, s_2, ..., s_m]$. We initialize the causal language model with $s$ and iteratively generate a content plan $y_i = [s_{m+1}, s_{m+2}, ..., <|EOS|>]$ until the end-of-sentence token is reached. By stopping at $<|EOS|>$, we obtain a single sentence that outlines the content needed for the next sentence of the expository document. If GPT-2 generates the end-of-text token, the document is completed.

### 3.2 Retriever

In order to effectively produce the information described in the content plan, we seek to narrow the search space of where these facts could occur. Thus, given a stylistic content plan $y_i$ produced by the Imitator, the Retriever $p(x|y_i)$ searches for the top-$k$ candidate facts $x \subseteq \mathcal{C}$ that contain the content described in $y_i$. We find that existing retrievers, such as DPR (Karpukhin et al., 2020) and BM25 (Robertson et al., 1995), may struggle to complete this task, as the hallucinated factual entities in the content plan $y_i$ impair these models' search capabilities. For example, when generating an expository text for *ML history*, the content plan $y_i$ may be the sentence "Machine learning was named by Noam Chomsky in 1984." The factual entities "Noam Chomsky" and "1984" should be ignored when searching for the correct facts, but DPR and BM25 still weigh these terms in their implementations.

To address this issue, we fine-tune DistilBERT (Sanh et al., 2019) with the task of classifying the index of each sentence of the expository texts in the training set (e.g. first sentence is labeled as 0). In doing so, DistilBERT gives lower token attribution scores for factual entities, as sentences from different expository texts with the same index will not share these entities (Analyzed in §5.5). DistilBERT performs fairly well on the classification task, given the consistent organization and style of expository texts in the training set.

We compute the relevance of each sentence $x_j \in \mathcal{C}$ to the content plan $y_i$ by taking the dot product of $x_j$ and $y_i$, both embedded by DistilBERT. To obtain these embeddings, we feed each sentence through the classifier and take its representation in the last layer, averaged over all tokens:

$$\mathbf{d}(x_j) = \text{DistilBERT}(x_j), \quad (2)$$

$$\mathbf{q}(y_i) = \text{DistilBERT}(y_i), \quad (3)$$

$$p(x_j|y_i) \sim \mathbf{d}(x_j)^T \mathbf{q}(y_i). \quad (4)$$

The top-$k$ most relevant factual sentences $x \subseteq \mathcal{C}$ to $y_i$ will have the $k$-highest values for $p(x_j|y_i)$, which can be obtained through Maximum Inner-Product Search (Shrivastava and Li, 2014).

### 3.3 Paraphraser

To ensure the expository document flows smoothly, we must reword the retrieved factual information in the style of the expository document domain. Thus, after obtaining a stylistic content plan $y_i$ and factual sentences $x$, the Paraphraser $p(z|x, y_i)$ must generate a single sentence $z$ aligned to the syntax of $y_i$ and the semantics of $x$. To achieve this goal, we formulate a variation on text generation with syntactic exemplars (Chen et al., 2019; Lin et al., 2020). We aim to minimize the cross-entropy loss of token prediction for $z$, conditioned on $y_i$ and $x$:

$$\lambda_{para} = -\sum_{k=1}^{|z|} \log p(z_k|y_i, x, z_1, ..., z_{k-1}). \quad (5)$$

We minimize $\lambda_{para}$ with BART (Lewis et al., 2020a), a seq2seq transformer-based language model. We modify the input so $x$ and $y_i$ are surrounded by custom $<|\texttt{fact}|>$ and $<|\texttt{style}|>$ tokens, respectively. To add additional context to the Paraphraser, we include the topic of the expository document $t$ surrounded by a custom token $<|\texttt{topic}|>$ in the input. Using the generated stylistic content plan $y_i$, retrieved facts $x$, and the topic document of the document $t$, we train BART to generate the the next sentence $z$ in the ground truth expository text.

Our problem formulation differs from traditional text generation with syntactic exemplars, in that the input $x$ contains multiple sentences instead of one. This change is necessary, as the information outlined in the content plan $y_i$ may be distributed across multiple sentences. Thus, BART must learn to synthesize information from multiple sentences while adhering to the style of the content plan.

### 3.4 The Iterative IRP Framework

The Imitator, Retriever, and Paraphraser are combined to generate expository documents, detailed in Algorithm 1. After the topic $t$, prefix $r$, and factual corpus $\mathcal{C}$ are provided, the Imitator first uses $r$ as the initial context for GPT-2 to generate a stylistic content plan $y_i$. Next, the Retriever embeds $y_i$ and each sentence $x_j \in \mathcal{C}$ with DistilBERT, in order to find the top-$k$ factual sentences $x \subseteq \mathcal{C}$ most similar

---

**Algorithm 1** Imitate, Retrieve, Paraphrase

1: **procedure** IRP($t, r, \mathcal{C}$)
2:     Initialize $\mathcal{D}$                       ▷ $\mathcal{D}$ is the output
3:     Initialize $p \leftarrow r$                     ▷ $p$ is the prefix
4:     **while** true **do**
5:         $y_i \leftarrow$ IMITATE($p$)
6:         **if** $y_i = <|\texttt{endoftext}|>$ **then**
7:             **return** $\mathcal{D}$
8:         $x \leftarrow$ RETRIEVE($y_i, \mathcal{C}$)
9:         $z \leftarrow$ PARAPHRASE($y_i, x, t$)
10:        Append $z$ to $\mathcal{D}$
11:        $p \leftarrow \mathcal{D}$

---

to $y_i$. Finally, the Paraphraser uses BART to combine the syntax of $y_i$ and the semantics of $x$ into a single sentence $z$, which is appended to the output $\mathcal{D}$. The next prefix for the Imitator is set to $\mathcal{D}$, and the process is repeated until the generated content plan $y_i$ is the $<|\texttt{endoftext}|>$ token.

### 3.5 Training

The Imitator, Retriever, and Paraphraser modules are trained independently to tackle expository text generation. We will now describe how we modify an expository text generation training set to train each of these components. The training set contains triples of titles (input), factual corpora (input), and expository texts (output), i.e., $(t, \mathcal{C}, \mathcal{D})$. The Imitator and Retriever are trained without modifying the training set, solely leveraging the expository texts. The Imitator performs causal language modeling with GPT-2 on each document $\mathcal{D}$, while the Retriever uses DistilBERT to classify the position of every sentence comprising each document $\mathcal{D}$.

To train the Paraphraser, we require triplets of stylistic content plans $y_i$, sets of factual sentences $x$, and reworded sentences $z$. For a given triplet, we can obtain the reworded sentence $z$ by selecting any of the sentences found in an expository document $\mathcal{D}$. Working backwards, we represent the stylistic content plan $y_i$ as a sentence from a different expository document that has high similarity to $z$, where similarity is calculated with Eq. 4. We obtain $x$ in a similar manner, using $z$ to retrieve the top-$k$ factual sentences $x \subseteq \mathcal{C}$, also according to Eq. 4. By using $z$ instead of $y_i$ to retrieve $x$, we can be more confident that $x$ will contain the information needed to reconstruct $z$, reducing the need for the Paraphraser to hallucinate during training.

## 4 Experimental Setup

### 4.1 Datasets

We test the capabilities of IRP on three diverse, newly-collected datasets. **1) U.S. News** is a corpus of 433 college descriptions from the top 500 ranked colleges on U.S. News.[2] We select the college name as the topic of the document. **2) Medline** contains information for 844 medications from MedlinePlus,[3] a medical library supported by the National Institute of Health. We select the medication name as the topic of the document. **3) WikiCS** is a collection of the first paragraphs of history sections from 500 Wikipedia[4] articles in computer science. We select the Wikipedia article title as the topic of the document. For each dataset, we create a 70/10/20 train/validation/test split. No data from the test set is used to train or validate any of the models or components of IRP. We provide full details for dataset collection in Appendix A.1.

As a preliminary study for expository text generation, we assume that the best corpus $\mathcal{C}$ has already been obtained for each document $\mathcal{D}$. This is an approximation for the scenario where $\mathcal{C}$ is acquired in real time. To obtain such ideal corpora, we collect documents from the web, reverse engineering with $\mathcal{D}$. We web scrape sentences $\mathcal{W}$ from the top-5 web pages returned using the topic $t$ and each sentence of $\mathcal{D}$ as search queries. We exclude pages that contain the ground truth text $\mathcal{D}$, such as any website with a URL containing "usnews" for U.S. News. In most cases, we find that the retrieved sentences $\mathcal{W}$ provide the necessary facts for generating $\mathcal{D}$ (§5.4). But to guarantee a dataset that provides all facts, we create two versions of each dataset, one where $\mathcal{C} = \mathcal{W}$ and one where $\mathcal{C} = \mathcal{W} \cup \mathcal{D}$, denoted by *without doc* and *with doc*, respectively. To introduce variation specifically for the *with doc* datasets, we perform back translation (Mallinson et al., 2017) on each expository text $\mathcal{D}$ and use it as the gold output, which we found not to affect its factual content and preserved its style (i.e. organization and phrasing). For the scope of this work, we assume that $\mathcal{C}$ contains accurate, consistent facts, and we believe future works could explore fact-checking (Rashkin et al., 2017) for a more robust expository text generation model. The corpora $\mathcal{C}$ are shuffled in each dataset.

---

[2] https://www.usnews.com/best-colleges
[3] https://medlineplus.gov/druginfo
[4] https://en.wikipedia.org/

### 4.2 Baselines

We compare IRP with the following baselines:
**1) LLaMA** (Touvron et al., 2023) is an LLM shown to have competitive performance with GPT-3. We choose the 7B version of LLaMA and prompt with 5 representative training examples. LLaMA prefixes its output with the same prefixes used by IRP.
**2) LLaMA+Retr** is LLaMA with an extra input of the top-5 retrieved sentences with DPR from the factual corpus using the document title as a query.
**3) LED** (Beltagy et al., 2020) is a seq2seq LM leveraging the Longformer model to encode and decode long documents. LED uses the topic and corpus as inputs to generate the expository text.
**4) RAG** (Lewis et al., 2020b) uses DPR (Karpukhin et al., 2020) to retrieve the top-$k$ facts from the input corpus using the document title as the query. Using the query and facts as inputs, we then train BART Large to generate the expository document.
**5) BART** is trained to generate the output using the topic as the sole input. This model helps us assess if other models use the factual corpus, or if they simply memorize the style of the expository text.

### 4.3 Training Setup

IRP uses GPT-2 Large, DistilBERT Base, and BART Large for the Imitator, Retriever, and Paraphraser. We use "[topic] is," "[topic] is used to treat," and "[topic] was first created" as the prefixes for U.S. News, Medline, and WikiCS. These were selected by assessing common prefixes in the training set. As a quality control check after generation, we filter sentences deemed repetitive by the Retriever embeddings (cosine similarity above 0.98). We discuss more training details in Appendix A.2.

### 4.4 Quantitative Metrics

We evaluate the quality of the generated documents with two sets of metrics. First, we use *traditional metrics*. ROUGE-1 (**R1**) and ROUGE-2 (**R2**) (Lin, 2004), **BLEU** (Papineni et al., 2002), and **METEOR** (Denkowski and Lavie, 2014) measure the similarity between the predicted and true outputs.

However, as these metrics have low correlations with human judgements of factuality (Kryscinski et al., 2020; Fabbri et al., 2022), we also adopt existing *factuality metrics*. First, we calculate the average percentage of tokens in the generated text that are **Halluc**inated, meaning that they do not appear in the input corpus. Next, we use **FactCC**, a classifier that is trained to detect factual errors

| Datasets | Models | Traditional Metrics | | | | Factuality Metrics | | | | Length |
|---|---|---|---|---|---|---|---|---|---|---|
| | | R1 | R2 | BLEU | METEOR | Halluc | FactCC | NLI-Ent | NLI-Contr | |
| U.S. News *with doc* | **IRP (Ours)** | **0.911\*** | **0.828\*** | **0.802\*** | **0.900\*** | **3.59\*** | **0.934\*** | **0.903\*** | **0.023\*** | 1.01 |
| | LLaMA | 0.757 | 0.601 | 0.540 | 0.749 | 7.90 | 0.609 | 0.385 | 0.547 | 0.91 |
| | LLaMA+Retr | 0.759 | 0.613 | 0.548 | 0.753 | 7.56 | 0.608 | 0.387 | 0.562 | 0.91 |
| | LED | 0.857 | 0.738 | 0.700 | 0.844 | 4.69 | 0.727 | 0.702 | 0.184 | 0.98 |
| | RAG | 0.788 | 0.651 | 0.611 | 0.821 | 7.40 | 0.475 | 0.402 | 0.515 | 1.07 |
| | BART | 0.774 | 0.621 | 0.586 | 0.807 | 9.94 | 0.341 | 0.255 | 0.682 | 1.04 |
| Medline *with doc* | **IRP (Ours)** | 0.551 | 0.435 | 0.318 | 0.490 | **1.71\*** | 0.871 | 0.521 | **0.126\*** | 0.66 |
| | LLaMA | 0.398 | 0.196 | 0.127 | 0.301 | 2.33 | 0.875 | 0.326 | 0.165 | 0.67 |
| | LLaMA+Retr | 0.484 | 0.282 | 0.206 | 0.403 | 2.89 | 0.853 | 0.368 | 0.181 | 0.77 |
| | LED | **0.671\*** | **0.549\*** | **0.453\*** | **0.611\*** | 2.32 | **0.933\*** | **0.571** | 0.249 | 0.68 |
| | RAG | 0.587 | 0.446 | 0.385 | 0.526 | 3.03 | 0.866 | 0.355 | 0.207 | 0.85 |
| | BART | 0.400 | 0.205 | 0.179 | 0.355 | 12.12 | 0.805 | 0.098 | 0.379 | 0.94 |
| WikiCS *with doc* | **IRP (Ours)** | **0.490\*** | **0.372\*** | **0.338\*** | **0.442\*** | **0.68** | 0.616 | **0.355** | **0.147** | 0.88 |
| | LLaMA | 0.166 | 0.025 | 0.010 | 0.151 | 9.37 | 0.366 | 0.057 | 0.525 | 1.92 |
| | LLaMA+Retr | 0.169 | 0.025 | 0.010 | 0.153 | 4.51 | **0.740** | 0.172 | 0.250 | 1.93 |
| | LED | 0.250 | 0.120 | 0.046 | 0.158 | 0.79 | 0.518 | 0.141 | 0.162 | 0.45 |
| | RAG | 0.327 | 0.201 | 0.076 | 0.228 | 1.11 | 0.567 | 0.326 | 0.170 | 0.42 |
| | BART | 0.231 | 0.047 | 0.009 | 0.129 | 11.22 | 0.390 | 0.145 | 0.340 | 0.43 |
| U.S. News *without doc* | **IRP (Ours)** | **0.807\*** | **0.675\*** | **0.649\*** | 0.816 | **10.61** | 0.609 | **0.470** | **0.437\*** | 1.01 |
| | LLaMA | 0.757 | 0.601 | 0.540 | 0.749 | 12.34 | **0.609** | 0.385 | 0.547 | 0.91 |
| | LLaMA+Retr | 0.759 | 0.612 | 0.547 | 0.753 | 11.86 | 0.602 | 0.382 | 0.567 | 0.91 |
| | LED | 0.792 | 0.624 | 0.813 | 0.776 | 10.71 | 0.539 | 0.468 | 0.544 | 1.09 |
| | RAG | 0.793 | 0.653 | 0.613 | **0.824** | 11.84 | 0.449 | 0.351 | 0.593 | 1.05 |
| | BART | 0.774 | 0.621 | 0.586 | 0.807 | 14.30 | 0.341 | 0.255 | 0.682 | 1.00 |
| Medline *without doc* | **IRP (Ours)** | 0.512 | 0.347 | 0.257 | 0.446 | **2.76** | **0.883** | 0.448 | **0.171** | 0.69 |
| | LLaMA | 0.398 | 0.196 | 0.127 | 0.301 | 3.11 | 0.875 | 0.326 | 0.165 | 0.67 |
| | LLaMA+Retr | 0.388 | 0.182 | 0.111 | 0.287 | 2.82 | 0.869 | 0.319 | 0.174 | 0.65 |
| | LED | 0.545 | **0.389** | 0.244 | 0.450 | 2.92 | 0.873 | **0.491** | 0.173 | 0.64 |
| | RAG | **0.548** | 0.369 | **0.324\*** | **0.504** | 4.12 | 0.819 | 0.356 | 0.213 | 0.96 |
| | BART | 0.400 | 0.205 | 0.179 | 0.355 | 12.50 | 0.805 | 0.098 | 0.379 | 0.94 |
| WikiCS *without doc* | **IRP (Ours)** | **0.380\*** | **0.209\*** | **0.159\*** | **0.305\*** | **2.43** | 0.491 | **0.263** | **0.193** | 0.82 |
| | LLaMA | 0.166 | 0.025 | 0.010 | 0.151 | 9.51 | 0.366 | 0.057 | 0.525 | 1.92 |
| | LLaMA+Retr | 0.172 | 0.026 | 0.010 | 0.151 | 4.64 | **0.736** | 0.178 | 0.257 | 1.89 |
| | LED | 0.261 | 0.107 | 0.033 | 0.169 | 2.94 | 0.377 | 0.160 | 0.302 | 0.42 |
| | RAG | 0.312 | 0.139 | 0.060 | 0.214 | 3.42 | 0.537 | 0.237 | 0.220 | 0.54 |
| | BART | 0.231 | 0.047 | 0.009 | 0.129 | 11.48 | 0.390 | 0.145 | 0.340 | 0.43 |

Table 1: Comparison of *traditional metrics* (ROUGE-1, ROUGE-2, BLEU, METEOR) and *factuality metrics* (Hallucinations, FactCC, Entailment, Contradictions) for expository text generation models. Models with values marked with * significantly outperform all baselines ($p < 0.05$, Wilcoxon signed-rank test (Woolson, 2007)).

between a source text and a claim (Kryscinski et al., 2020). We use the true output as the source and each sentence of the generated text as the claim, and report the average proportion of source/claim pairs predicted as factually consistent. Finally, as research has suggested that natural language inference has a high correlation with human judgements of factuality (Maynez et al., 2020b), we calculate if the generated text is entailed (**NLI-Ent**) or contradicted (**NLI-Contr**) by the true output. We train a DistilBERT classifier (accuracy of 0.82) on the MNLI dataset (Williams et al., 2018), and report the average proportion of generated sentences that are predicted to be entailed/contradicted by the true output. All metrics are reported from a single run.

# 5 Results

## 5.1 Performance Comparison

In Table 1, we observe that IRP obtains the highest factuality scores, achieving the strongest results for 19 out of the 24 calculated factuality metrics. Further, we find that apart from Medline, IRP outperforms baselines on almost all traditional metrics. These findings confirm that the Paraphraser faithfully rewords the retrieved facts in the style of the expository document domain. Previous works have shown that prioritizing factuality leads to a drop in ROUGE (Goodrich et al., 2019; Maynez et al., 2020b). However, IRP adheres to factual accuracy while maintaining the style of the expository document domain, suggesting the benefit of iteratively

| Dataset | Model | with doc | | | without doc | | |
|---|---|---|---|---|---|---|---|
| | | Style | Fact | Avg | Style | Fact | Avg |
| U.S. News | IRP (**Ours**) | **4.92** | **4.57** | **4.75** | **4.90** | 3.80 | 4.35 |
| | LLaMA+Retr | 4.82 | 3.30 | 4.06 | 4.77 | 3.25 | 4.01 |
| | LED | 4.65 | 3.78 | 4.22 | 4.42 | 2.88 | 3.65 |
| | RAG | 4.75 | 2.70 | 3.73 | 4.73 | 2.85 | 3.79 |
| | ChatGPT | 4.72 | 4.10 | 4.41 | 4.72 | **4.10** | **4.41** |
| Medline | IRP (**Ours**) | **4.53** | 4.78 | **4.66** | 4.12 | **4.88** | **4.50** |
| | LLaMA+Retr | 3.48 | 4.55 | 4.02 | 3.52 | 4.28 | 3.90 |
| | LED | 4.32 | 4.08 | 4.20 | **4.28** | 4.30 | 4.29 |
| | RAG | 4.28 | 4.15 | 4.22 | 4.10 | 4.13 | 4.12 |
| | ChatGPT | 3.53 | **4.83** | 4.18 | 3.53 | 4.83 | 4.18 |

Table 2: Human evaluation of style and factuality of expository documents on a 5-Likert scale. **Avg** is the average of style and factuality scores. ChatGPT is prompted with five representative training examples.

| Model | R1 | Halluc | FactCC | NLI-Ent | NLI-Contr |
|---|---|---|---|---|---|
| IRP Full | **0.490** | **0.68** | **0.616** | **0.355** | **0.147** |
| Topic Query | 0.165 | 2.58 | 0.329 | 0.206 | 0.229 |
| Gen All | 0.412 | 2.62 | 0.493 | 0.276 | 0.240 |

Table 3: R1 and factuality of IRP versus **Gen**erating text **All** at once instead of iteratively, and using a **Topic Query** over stylistic content plans on WikiCS *with doc*.

performing content planning, retrieval, and paraphrasing for expository text generation.

We also find that LLaMA+Retr does not consistently outperform LLaMA, implying that the LLM equipped with DPR cannot effectively search and use the factual corpus. Further, although RAG is typically effective in knowledge intensive tasks, the model obtains lower factuality scores than IRP in 23/24 metrics. Both findings suggest that the document title used by LLaMA+Retr and RAG is an insufficient query to retrieve the factual information needed to produce expository text. Hence, fine-grained queries, such as the stylistic content plans used by IRP, are a better strategy for obtaining all of the facts to include in the expository text.

Finally, we note that most models achieved much better performance on the *with doc* datasets. However, the *with doc* scenario is unrealistic, indicating that future models should prioritize the knowledge acquisition step of $\mathcal{C}$, as it largely dictates the factuality of the output. We believe that studying models that search the web during inference (e.g. LLMs with search engines) is a promising next step toward stronger expository text generation models.

## 5.2 Human Evaluation

To further test the quality of the generated expository texts, we invite two computer science and engineering college students to evaluate 30 random expository documents in the test sets produced by each baseline on U.S. News and Medline. Similar to previous works (Hua and Wang, 2019; Balachandran et al., 2022), we ask annotators to evaluate on **Style** adherence to the true output (i.e. organization and phrasing), and **Fact**uality on a 5-Likert scale. For Fact, we provide annotators with the true out-

put and encourage them to use external resources (Google Search). We observe high annotator agreement for Fact and Style, with Krippendorff's $\alpha$ (Krippendorff, 2011) around 0.70 on each dataset.

IRP strikes the best balance between factuality and style (**Avg**) in 3/4 datasets and competes with ChatGPT on the fourth dataset (Table 2), despite having less parameters (1.2B vs 175B). Generally, we note that the LLMs (ChatGPT, LLaMA+Retr) perform better in factuality but worse in style, while the opposite is true for seq2seq LMs (LED, RAG).

## 5.3 Ablation Study

We conduct an ablation study (Table 3, full results Table 6) and find that using stylistic content plans over the topic as a query and generating text at the sentence level instead of all at once, both improve the performance of IRP. This suggests that an iterative RAG model that creates fine-grained queries, such as IRP, is a preferred strategy for our task.

## 5.4 Factual Error Investigation

To study the errors produced by IRP, we invite one computer science student to annotate 30 expository texts produced by IRP on the *without doc* datasets. First, we ask the annotator to identify errors, i.e., facts in the generated text that do not exist in the true output. We then ask if each error occurred because 1) an **alternative** fact exists in the retrieved sentences (i.e. no hallucination), 2) **no** suitable **fact** could have been found by the Retriever, as it does not exist in the corpus, 3) the fact exists in the corpus, but the **Retriever** could not locate it, or 4) the Retriever located the correct fact, but the **Paraphraser** hallucinated. We store each step of IRP so the annotator can answer this question.

We find that many factual errors are due to alternatives existing in the retrieved facts, rather than the weaknesses of the Retriever, Paraphraser or data collection (Figure 4). For example, one source may report an outdated university ranking compared to

---

[5]https://github.com/cdpierse/transformers-interpret

| | |
|---|---|
| **U.S. News** | *Original*: Emory University is a private institution founded in 1836 |
| | emory university is a private institution founded in 1836 |
| **Medline** | *Original*: Hydroxyzine belongs to a class of medications called antihistamines |
| | hydro ##xy ##zine belongs to a class of medications called anti ##his ##tam ##ines |
| **WikiCS** | *Original*: The name Prologue was born around 1972 by Alain Colmerauer and Philippe Roussel |
| | the name prologue was born around 1972 by alain col ##mer ##auer and philippe ro ##uss ##el |

Figure 3: Visualized token attribution scores for the classification task performed by the Retriever on a sample of sentences from each test set. Darker shades of blue indicate higher token attribution scores. Scores are calculated with transformers-interpret[5], a Python library that leverages Captum (Kokhlikyan et al., 2020).

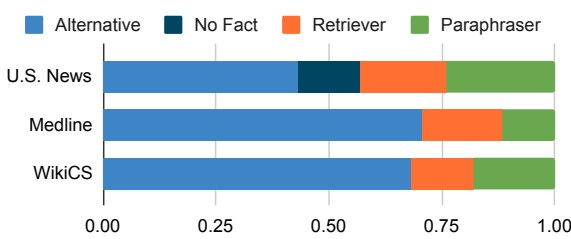

Figure 4: Distribution of IRP factual error types.

U.S. News. This poses an interesting question for future lines of work on expository text generation: how can we leverage fact verification to accurately select information when multiple options exist?

### 5.5 Retriever Embedding Analysis

The Retriever ignores hallucinated entities when creating embeddings, resulting in better search capabilities compared to pretrained retrievers. We visualize this property in Figure 3 and find that the Retriever puts less weight on factually specific entities (e.g. "Emory" and "Prologue"). Hence, the Retriever can focus its embeddings on the more important terms for retrieving information (e.g. "belongs," "class," and "medications"). We show the strength of the Retriever quantitatively in Table 5.

### 5.6 Sample Expository Document Outputs

We provide examples of documents generated by models on our three datasets in Appendix A.3.

## 6 Conclusion

We introduce the task of expository text generation and design IRP to overcome the limitations of existing models. IRP explicitly and iteratively performs content planning, fact selection, and paraphrasing to generate high quality expository texts. Automatic and human evaluations on three newly-collected datasets reveal that IRP preserves factual accuracy while maintaining stylistic consistency. Our ablation studies confirm the importance of sentence-level generation and creating fine-grained search queries. Finally, we visualize the Retriever of IRP and study the factual errors produced by our model to suggest future research directions.

## 7 Limitations

One drawback of IRP is that three separate components are trained for each expository document domain. For our initial exploration of expository text generation, we train three separate components, as it made our model more interpretable and it was thus simpler to detect errors in our components. The fact that IRP is not trained end-to-end suggests that there is still room for improvement and research in expository text generation.

In addition, IRP is computationally more expensive than traditional RAG models, as our runtime scales linearly with respect to the number of sentences in the output. Although IRP shows improvements over baselines, we acknowledge that it is important to ensure IRP is computationally efficient. To overcome this issue, we believe that future iterative RAG models could improve upon IRP by exploring efficient algorithms for maximum inner-product search, as well as developing a batched version of IRP that can generate multiple sentences in tandem.

Lastly, we find that expository text generation frameworks have a large performance gap between the *with doc* and *without doc* datasets (§5.1). As discussed in the paper, we believe this gap can be overcome by studying and developing models that can retrieve information from the web in real time during inference. For example, instead of using stylistic content plans to search a provided factual corpus, perhaps they could be reworded into search queries to retrieve up-to-date information

from Google in real time, thus overcoming any limitations of a provided factual corpus. If future work in this direction results in expository text generation models that can perform live retrieval during inference, they can also be compared and benchmarked with LLMs that are equipped with web search engine plugins.

# 8   Ethical Considerations

The fundamental goal of IRP is to generate factual content. However, as with all text generation frameworks, IRP may produce factual errors, as shown in §5.4. Future expository text generation models could ameliorate the harms of factual errors by performing fact verification, retrieving live information from the web during inference, incorporating more knowledge sources, or attributing the source of the generated facts for increased transparency.

Further, the Paraphraser is the key component in IRP that ensures the generated text is faithful to the factual corpus. However, there is always the possibility of someone creating their own Paraphraser aligned with the goal of producing misinformation or deceptive claims from true information and plugging this malicious component into IRP. We hope that future research will result in safeguards to detect and combat these potential risks of seq2seq language models.

# 9   Acknowledgements

We thank the anonymous reviewers for their feedback. This material is based upon work supported by the National Science Foundation IIS 16-19302 and IIS 16-33755, Zhejiang University ZJU Research 083650, IBM-Illinois Center for Cognitive Computing Systems Research (C3SR) - a research collaboration as part of the IBM Cognitive Horizon Network, grants from eBay and Microsoft Azure, UIUC OVCR CCIL Planning Grant 434S34, UIUC CSBS Small Grant 434C8U, and UIUC New Frontiers Initiative. Any opinions, findings, and conclusions or recommendations expressed in this publication are those of the author(s) and do not necessarily reflect the views of the funding agencies.

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

# A Appendix

## A.1 Detailed Dataset Collection

To obtain the expository documents $\mathcal{D}$ on each dataset, we web scrape the respective websites with BeautifulSoup[6]. We could not find specific research licenses for the three datasets, but note that they are free to access and publicly available online. Further, we found that each dataset has been analyzed in previous NLP research papers.

For a given expository document $\mathcal{D}$ and its topic $t$, we will now explain how we obtain the set of factual sentences $\mathcal{W}$, briefly described in §4.1. First, we break up $\mathcal{D}$ into a set of sentences $\{y_i\}$. For each sentence $y_i$, we obtain the URLs of the top 5 search results using the query "$[t]$ $[y_i]$". After repeating this for each sentence, we flatten the list of URLs into a unique set, and filter the URLs that contain the ground truth expository document (e.g. for the U.S. News dataset, we filter all URLs which contain the substring "usnews"). We then use BeautifulSoup to obtain the text of all of the `<p>` tags. Using the nltk sentence tokenizer[7], we extract all sentences and flatten them into a unique set.

We clean sentences by keeping alpha-numeric symbols and punctuation with regex, as well as applying unidecode[8] to ensure sentences contain only ASCII characters. All information is in English, and we studied a sample of sentences to ensure that there was no offensive language in the dataset. By analyzing a sample of the dataset, we did not find any personal identifiable information (PII), but to be cautious, we use the Presidio[9] analyzer provided by Microsoft and remove all sentences with detected PII (prediction score > 0.3), encompassing the following entities: "PHONE NUMBER", "CRYPTO", "EMAIL ADDRESS", "IBAN CODE", "IP ADDRESS", "MEDICAL LICENSE", "US BANK NUMBER", "US DRIVER LICENSE", "US ITIN", "US PASSPORT", "US SSN". In Table 4, we display summary statistics of each dataset after this process.

---

[6]https://pypi.org/project/beautifulsoup4/
[7]https://www.nltk.org/api/nltk.tokenize.html
[8]https://pypi.org/project/Unidecode/
[9]https://microsoft.github.io/presidio/analyzer/

## A.2 Detailed Training Setup

The Imitator is trained with GPT-2 Large (774M parameters) through the aitextgen[10] Python package. We choose a batch size of 1, a learning rate of 1e-3, and train the model for 3000 steps. All other parameters are set to the default value of the aitextgen implementation. The Retriever is trained with DistilBERT Base uncased (66M parameters). We choose a batch size of 16, a learning rate of 2e-5, a weight decay of 0.01, and 1 training epoch. The Paraphraser is trained with BART Large (406M parameters). We choose a batch size of 32, a learning rate of 2e-5, a weight decay of 0.01, a gradient accumulation step size of 8, and 5 training epochs. The Paraphraser takes $\sim$2 hours to finish training. Each component of IRP is optimized with the AdamW optimizer and trained with a single NVIDIA A40 GPU. The parameters, resources, and models remain the same for each dataset, with only slight differences in training time. During prediction, GPT-2 uses a temperature of 0.7 and generates text with a maximum length of 512, DistilBERT retrieves $k = 15$ factual sentences, and BART generates text with a maximum input and output length of 512.

LED uses an input size of 16384 and is trained with a batch size of 8, a learning rate of 5e-5, 8 gradient accumulation steps, 1500 warm-up steps, and trained for 8 epochs. The generator of RAG and the BART baseline are trained with the same parameters as the Paraphraser. The retriever of RAG selects $k = 15$ sentences. All unstated parameters are the default values of their respective implementations. We ensure that each model is trained until the validation loss converges.

For the LLaMA and LLaMA+Retr, we perform 5-shot prompting using 5 manually selected, representative input/output training examples, an example of which is shown in Figure 5. For ChatGPT, we use the web interface[11] and perform 5-shot prompting. We assess the outputs of all baselines and perform the same quality control check as IRP to filter semantically repetitive sentences, improving fluency.

## A.3 Qualitative Analysis

In Tables 7, 8, and 9, we present examples of expository documents generated by IRP on U.S. News, Medline, and WikiCS, respectively, on both ver-

---

[10]https://docs.aitextgen.io/
[11]https://chat.openai.com/

sions of the datasets (*with doc* and *without doc*). We also display the topic of the expository document and the true output. In these examples, we can see that IRP produces text with high factual accuracy without sacrificing fluency or the style of the expository document domain.

Further, in Tables 10, 11, and 12, we directly compare the expository document outputs of IRP and the baselines (LED, RAG, LLaMA, LLaMA+Retr) on U.S. News, Medline, and WikiCS, respectively. On U.S. News, we find that the baselines tend to produce factual errors related to many of the key details, such as the institution's founding and tuition. On Medline, we find that the baselines struggle to generate accurate drug classes and explanations for how the medications affect the human body. Some generated documents also contain phrases that are repetitive and difficult to understand. On WikiCS, we find that the baselines are mostly factually accurate, but the documents lack overall structure and coherence. Compared to other models, LLaMA struggles the most with preserving the style of the expository document domain.

### A.4 Human Evaluation

We display the set of instructions given to human annotators for evaluating the style and factual accuracy of expository documents in Figure 6.

```
title: { cache line }
output: { Early examples of CPU caches are the Atlas 2 and the IBM System / 360 Model 85 in the
1960s. The first CPUs to use a cache had only one cache level; unlike later Level 1 caches, it was not
split into L1d (for data) and L1i (for instructions). L1 cache started in 1976 with the IBM 801 CPU, became
mainstream in the late 1980s and entered the CPU market with the ARMv5TE in 1997. In 2015, the L1
cache was even split by sub-dollar SoCs. In addition, there are L2 caches there, and for larger processors
there are also L3 caches. The L2 cache is usually not shared and functions as a common repository for
the already shared L1 cache. Each core of a multi-core processor has a dedicated L1 cache and is not
normally shared between the CPUs. The cores can be split between caches located higher and higher. }
---
title: { exception handling }
output: { The first hardware exception handling was found in UNIVAC I in 1951. Arithmetic overflow
executed two commands at address 0 that could transfer control or correct the result. }
---
title: { functional programming }
output: { The lambda calculation, developed by Alonzo Church in the 1930s, is a formal calculation
system based on functional applications. In 1937, Alan Turing proved that the lambda calculation and
Turing machines are equivalent mathematical models and showed that the lambda calculation is
Turing-complete. Lambda calculation forms the basis of all functional programming languages. A
corresponding theoretical formulation, the combinatorial logic, was developed in the 1920s and 1930s by
Moses Schönfinkel and Haskell Curry. }
---
title: { variational inequalities }
output: { The first problem with varying inequality was the Signorini problem raised by Antonio Signorini
in 1959 and solved by Gaetano Fichera in 1963 after the references (Antman 1983, pp. 282-284) and
(Fichera 1995): the first works of the theory were (Fichera 1963) and (Fichera 1964a), (Fichera 1964b).
Later Guido Stampacchia proved his generalization of the Lax-Milgram theorem in (Stampacchia 1964) to
investigate the regularity problem for partial differential equations and coined the name "Inequality of
Variation" for all problems of this kind. Georges Duvaut encouraged his doctoral students to study and
expand Fichera's work after attending a conference in Brixen in 1965 at which Fichera presented his
study on the Signorini problem, as reported by Antman in 1983, p. 283: thus the theory became widely
known throughout France. }
---
title: { quantum computing }
output: { Quantum computing research began in 1980, when physicist Paul Benioff proposed a quantum
mechanical model of the Turing machine. Richard Feynman and Yuri Manin later suggested that a
quantum computer had the potential to simulate things that a classical computer could not do. In 1986,
Feynman introduced an early version of quantum circuit notation. In 1994, Peter Shor developed a
quantum algorithm to find the primary factors of an integrator with the potential to decrypt RSA-encrypted
communications. In 1998, Isaac Chuang, Neil Gershenfeld and Mark Kubinec created the first quantum
computer with two quantum bits to perform calculations. Despite ongoing experimental progress since the
late 1990s, most researchers believe that "fault-tolerant quantum computing is still a rather distant
dream." In recent years, investment in quantum computer research in the public and private sectors has
increased. On October 23, 2019, Google AI in partnership with NASA's National Space Administration
(NASA) is still a classical topic. }
---
title: { collective intelligence }
output: { Collective Intelligence was first created
```

Figure 5: Prompt given for LLaMA (no retrieval) on WikiCS. For LLaMA+Retr, we add "input: [information]" to the each in-context learning example, where information is the concatenated facts retrieved from DPR.

| Dataset | Train / Valid / Test | Average Number of Facts *with doc / without doc* | Average Output Length (words) |
|---------|----------------------|--------------------------------------------------|-------------------------------|
| U.S. News | 315 / 39 / 79 | 521.81 / 517.13 | 73.38 |
| Medline | 590 / 85 / 169 | 1002.89 / 999.47 | 84.31 |
| WikiCS | 350 / 50 / 100 | 973.58 / 970.58 | 90.33 |

Table 4: Summary statistics of U.S. News, Medline, and WikiCS datasets for expository document generation.

| | with doc | | | without doc | | |
|---|---|---|---|---|---|---|
| Model | U.S. News | Medline | WikiCS | U.S. News | Medline | WikiCS |
| IRP-DistilBERT | **0.665** | **0.926** | **0.703** | 0.578 | **0.771** | **0.580** |
| IRP-DPR | 0.643 | 0.795 | 0.573 | **0.642** | 0.716 | 0.573 |

Table 5: IRP Retriever comparison using our DistilBERT setup versus DPR as the Retriever. We run both IRP variations on each dataset where the Retriever obtains the top-5 factual sentences at each step. We store these factual sentences and calculate the average ROUGE-1 recall performance between all factual sentences and the ground truth output, which indicates the proportion of tokens in the output that were covered by the retrieved factual sentences. On 5/6 datasets, our DistilBERT retriever outperforms DPR (best results in bold).

| | | Traditional Metrics | | | | | Factuality Metrics | | |
|---|---|---|---|---|---|---|---|---|---|
| Dataset | Model | R1 | R2 | BLEU | METEOR | Halluc | FactCC | NLI-Ent | NLI-Contr |
| U.S. News *with doc* | IRP Full | **0.911** | **0.828** | **0.802** | **0.900** | **3.59** | **0.934** | **0.903** | **0.023** |
| | Gen All | 0.764 | 0.611 | 0.592 | 0.793 | 8.51 | 0.503 | 0.394 | 0.502 |
| | Topic Query | 0.301 | 0.170 | 0.132 | 0.320 | 3.64 | 0.307 | 0.599 | 0.337 |
| Medline *with doc* | IRP Full | **0.551** | **0.435** | **0.318** | 0.490 | **1.71** | **0.871** | **0.521** | **0.126** |
| | Gen All | 0.534 | 0.368 | **0.318** | **0.501** | 4.84 | 0.145 | 0.378 | 0.168 |
| | Topic Query | 0.281 | 0.141 | 0.137 | 0.244 | 3.92 | 0.066 | 0.330 | 0.156 |
| WikiCS *with doc* | IRP Full | **0.490** | **0.372** | **0.338** | **0.442** | **0.68** | **0.616** | **0.355** | **0.147** |
| | Gen All | 0.412 | 0.269 | 0.227 | 0.331 | 2.62 | 0.493 | 0.276 | 0.240 |
| | Topic Query | 0.165 | 0.067 | 0.022 | 0.113 | 2.58 | 0.329 | 0.206 | 0.229 |

Table 6: Comparison of *traditional metrics* (ROUGE-1, ROUGE-2, BLEU, METEOR) and *factuality metrics* (Hallucinations, FactCC, Entailment, Contradictions) for IRP ablations. **Gen All** generates the text at once rather than sentence-by-sentence. **Topic Query** uses the topic as the search query in the factual corpus instead of the stylistic content plans used by IRP. Best results are in bold.

| University (Topic) | True Output | IRP With Doc | IRP Without Doc |
|---|---|---|---|
| Stanford University | Stanford University is a private institution founded in 1885. It has a total of 7,645 students (fall 2021), its location is suburban, and the campus is 8,180 acres. It uses a quarterly academic calendar. Stanford University ranks 3rd in the 2022-2023 edition of the best colleges. Its tuition and fees are $56,169. | Stanford University is a private institution founded in 1885. It has a total of 7,645 students (fall 2021), its location is suburban, and the campus is 8,180 acres. It uses a quarterly academic calendar. Stanford University ranks 3rd in the 2022-2023 edition of the best colleges. Its tuition and fees are $56,169. | Stanford University is a private institution founded in 1885. It has a total of 7,645 students (fall 2021), its location is suburban, and the campus is 8,180 acres. It uses a quarterly academic calendar. Stanford University ranks 3rd in the 2022-2023 edition of the best colleges. Its tuition and fees are $56,169. |
| Grand Valley State University | Grand Valley State University is a public institution founded in 1960. It has a total of 19,379 students (fall 2021), its location is suburban, and the size of the campus is 1,391 acres. It uses a semester-based academic calendar. Grand Valley State University ranks 263rd among the top colleges in the 2022-2023 edition of the National Universities. Its tuition and fees in the states are $13,952; out-of-state tuition and fees are $19,856. | Grand Valley State University is a public institution founded in 1960. It has a total of 19,379 students (fall 2021), its location is suburban, and the campus is 1,391 acres. It uses a semester-based academic calendar. Grand Valley State University ranks 263rd among the top colleges in the 2022-2023 edition of the National Universities. Tuition and fees in the states are $13,952; out-of-state tuition and fees are $19,856. | Grand Valley State University is a public institution founded in 1960. It has a total of 16,851 students (fall 2021), a suburban location, and a 1,338-acre campus. It uses a semester-based academic calendar. Grand Valley State University ranks 263rd among the top colleges in the 2022-2023 edition of the National University Ranking. Tuition and fees are $11,952 in the states and $19,856 outside the states. |
| Southeastern University | Southeastern University is a private institution founded in 1935. It has a total of 8,118 students (fall 2021), a suburban location, and a campus of 88 acres. It uses a semester-based academic calendar. Southeastern University ranks # 331-440 in the 2022-2023 edition of the Top Colleges of the National Universities, with tuition and fees totaling $30,148. | Southeastern University is a private institution founded in 1935. It has a total of 8,118 students (fall 2021), its location is suburban, and the campus covers 88 acres. It uses a semester-based academic calendar. Southeastern University ranks # 331-440 among the top colleges in the 2022-2023 edition of the National University Ranking. Tuition and fees are $30,148. | Southeastern University is a private institution founded in 1935. It has a total of 8,118 students (fall 2021), its location is suburban, and the campus covers 88 acres. It uses a semester-based academic calendar. Southeastern University ranks 219th among the top colleges in the 2022-2023 edition of the National Universities. Its tuition and fees are $31,150. |
| University of Florida | The University of Florida is a public institution founded in 1853. It has a total of 34,881 enrolled students (fall 2021), its location is suburban, and the campus is 2,000 acres. It uses a semester-based academic calendar. The University of Florida ranks 29th in the rankings of the best colleges for the years 2022-2023. Tuition and fees in the states are $6,380; outside the state, tuition and fees are $28,658. | The University of Florida is a public institution founded in 1853. It has a total of 34,881 students (fall 2021), its location is suburban, and the campus is 2,000 acres. It uses a semester-based academic calendar. The University of Florida ranks 29th among the top colleges in the 2022-2023 edition. Tuition and fees in the states are $6,380; out-of-state tuition and fees are $28,658. | The University of Florida is a public institution founded in 1853. It has a total of 34,881 students (fall 2021), its location is suburban, and the campus covers 2,000 acres. It uses a semester-based academic calendar. The University of Florida ranks 29th among the top colleges in the 2022-2023 edition. Tuition and fees in the states are $6,381; out-of-state tuition and fees are $28,658. |

Table 7: Sample expository documents generated by IRP on U.S. News.

| Medicine (Topic) | True Output | IRP With Doc | IRP Without Doc |
|---|---|---|---|
| Pentazocine | Pentazocine is used to relieve moderate to severe pain. Pentazocine belongs to a class of drugs called opioid (narcotic) painkillers. It works by changing the way the brain and nervous system react to pain. | Pentazocine is used to relieve moderate to severe pain. Pentazocine belongs to a class of drugs called opioid (narcotic) painkillers. It works by changing the way the brain and nervous system react to pain. | Pentazocine is used to relieve moderate to severe pain. Pentazocine is only used to treat severe pain. Pentazocine belongs to a class of drugs called opioid (narcotic) painkillers. It works by blocking pain signals in the brain. |
| Thiothixene | Thiothixen is used to treat symptoms of schizophrenia (a mental illness that causes disturbed or unusual thinking, loss of interest in life, and strong or inappropriate emotions). Thiothixen belongs to a group of medications called conventional antipsychotics. It works by reducing abnormal arousal in the brain. | Thiothixene is used to treat symptoms of schizophrenia (a mental illness that causes disturbed or unusual thinking, loss of interest in life and strong or inappropriate emotions) in adults and children 12 years of age and older. Thiothixen belongs to a class of medications called conventional antipsychotics. It works by reducing abnormal arousal in the brain. | Thiothixene is used to treat symptoms of schizophrenia (a mental illness that causes disturbed thinking, loss of interest in life and strong or inappropriate emotions). Thiothixene belongs to a class of drugs called conventional antipsychotics. It works by reducing abnormal arousal in the brain. |
| Diltiazem | Diltiazem is used to treat high blood pressure and control angina pectoris (chest pain). Diltiazem belongs to a class of medications called calcium channel blockers. It relaxes blood vessels so the heart does not have to pump as hard. It also increases blood and oxygen supply to the heart. | Diltiazem is used to treat high blood pressure and angina pectoris (a condition in which the heart is unable to pump enough blood to all parts of the body). Diltiazem belongs to a class of medications called calcium channel blockers. It works by relaxing blood vessels in the body and heart and lowering the heart rate. | Diltiazem is used to treat certain types of heart rhythm disorders such as atrial fibrillation (a condition in which the heart beats irregularly, causing shortness of breath, dizziness and fatigue). Diltiazem belongs to a class of medications called calcium channel blockers. It works by relaxing blood vessels so that blood can flow to the heart. |
| Midostaurin | Midostaurin is used with other chemotherapy drugs to treat certain types of acute myeloid leukemia (AML; a type of cancer of the white blood cells). Midostaurin is also used for certain types of mastocytosis (a blood disease in which there are too many mast cells [a certain type of white blood cells]). Midostaurin belongs to a class of drugs called kinase inhibitors. It works by blocking the action of the abnormal protein that signals the proliferation of cancer cells, helping to stop the spread of mast and cancer cells. | Midostaurin is used with other chemotherapy drugs to treat certain types of acute myeloid leukemia (AML; a type of cancer of the white blood cells). Midostaurin is also used to treat certain types of mastocytosis (a blood disease in which there are too many mast cells [a certain type of white blood cells]). Midostaurin belongs to a class of drugs called kinase inhibitors. It blocks the action of the abnormal protein that signals cancer cells to multiply, helping to stop the spread of mast and cancer cells. | Midostaurin is used in combination with other medicines to treat a certain type of acute myeloid leukemia (AML; a cancer of the white blood cells) that has spread to other parts of the body or cannot be treated surgically. Midostaurin belongs to a class of drugs called kinase inhibitors. It works by blocking the action of the abnormal protein that signals cancer cells to multiply, helping to stop or slow the spread of cancer cells. |

Table 8: Sample expository documents generated by IRP on Medline.

| Wiki Title (Topic) | True Output | IRP With Doc | IRP Without Doc |
|---|---|---|---|
| Social Network Analysis | The analysis of social networks has its theoretical roots in the work of early sociologists such as Georg Simmel and Émile Durkheim, who wrote about the importance of studying the relationship patterns that bind social actors. Social scientists have been using the concept of "social networks" since the beginning of the 20th century to describe complex relationships between members of social systems at all levels, from interpersonal to international. | Social network analysis has its theoretical roots in the work of early sociologists such as Georg Simmel and Emile Durkheim, who wrote about the importance of studying the relationships that bind social actors and the need to organize social networks to ensure social stability. | Social network analysis has its theoretical roots in the work of early sociologists such as Georg Simmel and mile Durkheim, who wrote about the importance of investigating patterns of relationships that connect social actors. Social network analysis is not only a methodological approach to the analysis of relationships or structures, but also a theoretical method of structural analysis, which is widely used. |
| Cryptography | Before modern times, cryptography focused on message secrecy (i.e., encryption) - converting messages from an understandable form to an incomprehensible one and back again at the other end, rendering them unreadable to interceptors or eavesdroppers without secret knowledge (namely, the key needed to decrypt that message). Encryption sought to ensure secrecy in communications, for example, by spies, military leaders, and diplomats. In recent decades, the field has expanded beyond concerns about confidentiality, including techniques for verifying the integrity of messages, authenticating the identities of senders and recipients, digital signatures, interactive evidence, and secure calculations. | Essentially, before the early twentieth century, cryptography was primarily concerned with linguistic patterns. Other cryptographic primitives include the encryption algorithms themselves, one-way permutations, trapdoor permutations etc. Valuable information can be extracted from the traffic, such as the exchange of session tokens, which allows the attacker to access application accounts that the attacker should not be able to access. | Essentially, before the early twentieth century, cryptography was primarily concerned with the study of linguistic patterns. Other cryptographic primitives include the encryption algorithms themselves, one-way permutations, trapdoor permutation, etc. The security of many modern cryptographic techniques is based on the difficulty of certain computational problems, such as the integer factorization problem or the discrete logarithm problem. |
| Business Process Modeling | Business process modeling techniques such as flowchart, functional flowblock chart, control flowchart, Gantt chart, PERT chart, and IDEF have been around since the beginning of the 20th century. Gantt charts were among the first to appear around 1899, flowchart in the 1920s, functional flowblock chart, and PERT in the 1950s, data flowchart, and IDEF in the 1970s. Modern methods include Unified Modeling Language and Business Process Model and Notation, but they are only a fraction of the methods used over the years to document business processes. S. Williams coined the term "business process modeling" in his 1967 article "Business Process Modelling Improves Administrative Control." | S. Williams coined the term "business process modeling" in the 1960s in his 1967 article "Business Process Modelling Improves Administrative Control." These process models help organizations document workflows, surface key metrics, pinpoint potential problems and intelligently automate processes. Business process modeling became the foundation of new methods, for example those that supported data collection, data flow analysis, process flow diagrams and reporting facilities. Business process modeling techniques have been developed over the decades to support specific business needs. | The term "business process modeling" itself was coined in the 1960s in the field of systems engineering by S. Williams in his 1967 article "Business Process Modelling Improves Administrative Control." His idea was that techniques for obtaining a better understanding of physical control systems could be used in a similar way for business processes. It was not until the 1990s that the term became popular. |

Table 9: Sample expository documents generated by IRP on WikiCS.

In the following survey, you will read generated college descriptions in the style of U.S. News. Please rate each document on a scale of 1-5 in **Style Adherence,** and **Factual Accuracy**. Please use the following guidelines for these two attributes using one sentence from the example of the fictitious "Moon University."

**Style Adherence:** How similar is the generated text compared to the true output? Do they organize the same information in the same order, generally using the same phrasing? We are not concerned whether the factual information is correct, but rather if the factual information is being described/outlined appropriately.

Example True Output:

Moon University is a public institution that was founded in 2022.

Example Ratings:
**1** - Moon University has a total enrollment of 50 students.

**3** - Moon University was founded in 2022, is a public university, and enrolls 50 students.

**5** - Moon University is a public institution that was founded in 2005.

**Factual Accuracy:** How accurate is the information conveyed in the document? Are there significant factual inconsistencies or errors? We will provide the ground truth output along with the generated document. Please use Google to verify factual errors if they are not obvious.

Example True Output:

Moon University is a public institution that was founded in 2022. It has a total enrollment of 50.

Example Ratings:
**1** - Earth University is a private institution that was founded in 1990. It has 10,000 students.

**3** - Moon University is a public institution founded in 2000. It has a total enrollment of 75.

**5** - Moon University, founded in 2022, is a public institution. A total of 50 students are enrolled.

Figure 6: Evaluation instructions for expository text generation on U.S. News.

| Model | With Doc | Without Doc |
|---|---|---|
| True Output | Southeastern University is a private institution founded in 1935. It has a total of 8,118 students (fall 2021), a suburban location, and a campus of 88 acres. It uses a semester-based academic calendar. Southeastern University ranks # 331-440 in the 2022-2023 edition of the Top Colleges of the National Universities, with tuition and fees totaling $30,148. | Southeastern University is a private institution founded in 1935. It has a total of 8,118 students (fall 2021), a suburban location, and a campus of 88 acres. It uses a semester-based academic calendar. Southeastern University ranks # 331-440 in the 2022-2023 edition of the Top Colleges of the National Universities, with tuition and fees totaling $30,148. |
| **IRP (Ours)** | Southeastern University is a private institution founded in 1935. It has a total of 8,118 students (fall 2021), its location is suburban, and the campus covers 88 acres. It uses a semester-based academic calendar. Southeastern University ranks # 331-440 among the top colleges in the 2022-2023 edition of the National University Ranking. Tuition and fees are $30,148. | Southeastern University is a private institution founded in 1935. It has a total of 8,118 students (fall 2021), its location is suburban, and the campus covers 88 acres. It uses a semester-based academic calendar. Southeastern University ranks **219th** among the top colleges in the 2022-2023 edition of the National Universities. Its tuition and fees are $31,150. |
| LLaMa | Southeastern University is a private institution founded in 1935. It has a total of **2,500** students (fall 2021), its location is **urban**, and the campus is 100 acres in size. It uses a semester-based academic calendar. Southeastern University ranks **14th** in the 2022-2023 edition of the best colleges. Its tuition and fees are **$22,900**. | Southeastern University is a private institution founded in 1935. It has a total of **2,200** students (fall 2021), its location is **urban**, and the campus is 100 acres in size. It uses a semester-based academic calendar. Southeastern University ranks **14th** in the 2022-2023 edition of the best colleges. Its tuition and fees are **$22,900**. |
| LLaMa+Retr | Southeastern University is a private institution founded in 1935. It has a total of **1,800** students (fall 2021), its location is suburban, and the campus is 100 acres in size. It uses a semester-based academic calendar. Southeastern University ranks **12th** in the 2022-2023 edition of the best colleges. Its tuition and fees are **$21,000**. | Southeastern University is a private institution founded in 1935. It has a total of **1,900** students (fall 2021), its location is suburban, and the campus is 100 acres in size. It uses a semester-based academic calendar. Southeastern University ranks **12th** in the 2022-2023 edition of the best colleges. Its tuition and fees are **$21,000**. |
| LED | Founded in 1935, **Nova Southeastern University** is a private institution with a total of 8,118 students (fall 2021), a suburb of Lakeland, and a campus size of **330** acres. It uses a semester-based academic calendar. **Nova Southeastern University** ranks #331-440 among the top colleges in the 2022-2023 edition of the National Universities. Its tuition and fees are $30,148. | Founded in 1935, **Nova Southeastern University** is a private institution with a total of 8,118 students (fall 2021), a suburban location, and a campus size of 88 acres. It uses a semester-based academic calendar. **Nova Southeastern University** ranks **234th** among the top colleges in the 2022-2023 edition of the National Universities. Its tuition and fees are $30,148. |
| RAG | Southeastern University is a private institution founded in 1935. It has a total of 8,118 students (fall 2021), is located in the **city**, and the campus covers **250** acres. It uses a semester-based academic calendar. Southeastern University ranks #331-440 among the top colleges in the 2022-2023 edition of the National Universities. Tuition and fees are $32,834. | Southeastern University is a private institution founded in 1935. It has a total of 8,118 students (fall 2021), is located in the **city**, and the campus covers **250** acres. It uses a semester-based academic calendar. Southeastern University ranks #331-440 among the top colleges in the 2022-2023 edition of the National Universities. Tuition and fees are $32,180. |
| BART | Southeastern University is a private institution founded in **1891**. It has a total of **2,856** students (fall 2021), its location is suburban, and the campus covers 110 acres. It uses a semester-based academic calendar. Southeastern University ranks #331-440 among the top colleges in the 2022-2023 edition of the National Universities. Its tuition and fees are **$39,834**. | Southeastern University is a private institution founded in **1891**. It has a total of **2,856** students (fall 2021), its location is suburban, and the campus covers 110 acres. It uses a semester-based academic calendar. Southeastern University ranks #331-440 among the top colleges in the 2022-2023 edition of the National Universities. Its tuition and fees are **$39,834**. |

Table 10: Baseline comparison of generated expository documents on U.S. News for Southeastern University. Red text indicates factual errors.

| Model | With Doc | Without Doc |
|---|---|---|
| True Output | Daridorexant belongs to a class of drugs called orexin receptor antagonists. It works by blocking the action of a certain natural substance in the brain that causes alertness. | Daridorexant belongs to a class of drugs called orexin receptor antagonists. It works by blocking the action of a certain natural substance in the brain that causes alertness. |
| **IRP (Ours)** | Daridorexant is used to treat symptoms of pulmonary embolism (PE; a cancer in which the lungs are damaged) in people who have not been treated with other medications. Daridorexant belongs to a class of drugs called orexin receptor antagonists. It works by blocking the action of a certain natural substance in the brain that causes alertness. | Daridorexant belongs to a class of medications called dual orexin receptor antagonists (DORAs). It blocks the binding of the wake-promoting neuropeptide orexin and can turn off overactive wakefulness. |
| LLaMa | Daridorexant is used to treat insomnia (difficulty falling asleep or staying asleep). **It works by blocking the action of histamine, a substance in the body that can cause drowsiness**. | Daridorexant is used to treat insomnia (difficulty falling asleep or staying asleep). **It works by blocking the action of histamine, a substance in the body that can cause drowsiness**. |
| LLaMa+Retr | Daridorexant is a drug which acts as an antagonist of orexin receptors. It is used for the treatment of insomnia. | Daridorexant is used to treat insomnia (difficulty falling asleep or staying asleep). **It works by blocking the action of histamine, a substance in the body that can cause drowsiness**. |
| LED | Daridorexant is used to treat insomnia (**insomnia**; a condition in which disturbed sleep is difficult to control and can lead to difficulty sleeping and poor daytime functioning). Daridorexant belongs to a class of drugs called orexin receptor antagonists. **It works by reducing the amount of sleep-promoting substances in the brain**. Daridorexant does not cause insomnia **or improve daytime functioning**. | Daridorexant is used to treat insomnia (**insomnia**; a sleep disorder in which sleep is disturbed and difficult to control) in adults and children 12 years of age and older. Daridorexant belongs to a class of drugs called orexin receptor antagonists. It works by slowing activity in the brain to enable sleep. |
| RAG | Daridorexant is used alone or in combination with other medications to treat insomnia (**insomnia** or disturbed sleep) in adults and children **6 years of age and older**. It works by slowing activity in the brain to enable sleep. | Daridorexant is used alone or in combination with other medications to treat insomnia (**insomnia** or disturbed sleep). It is also used to treat restless leg syndrome (RLS; restless legs syndrome; a condition in which the legs do not function properly and the person can experience weakness, numbness, loss of muscle coordination and problems falling asleep) in adults and children **6 years of age and older** who have not been successfully treated with benzodiazepines (**narcotic**) or sedatives (**sudden, unexpected bouts of insomnia that can occur when the person is not treated with these medications**). |
| BART | Daridorexant is used alone or in combination with other medications to treat **high blood pressure**. It is also used to treat **agina pectoris** (chest pain that occurs when the heart does not pump enough blood to all parts of the body) in adults and children **2 years of age and older**. Daridorezant belongs to a class of medications called **angiotensin II receptor antagonists**. It blocks the action of certain natural substances that **tighten blood vessels**, allowing blood to flow more smoothly through the body. | Daridorexant is used alone or in combination with other medications to treat **high blood pressure**. It is also used to treat **agina pectoris** (chest pain that occurs when the heart does not pump enough blood to all parts of the body) in adults and children **2 years of age and older**. Daridorezant belongs to a class of medications called **angiotensin II receptor antagonists**. It blocks the action of certain natural substances that **tighten blood vessels**, allowing blood to flow more smoothly through the body. |

Table 11: Baseline comparison of generated expository documents on Medline for Daridorexant. Red text indicates factual errors. Blue text indicates repetitive or difficult to understand phrasing.

| Model | With Doc | Without Doc |
|---|---|---|
| True Output | Business process modeling techniques such as flowchart, functional flowblock chart, control flowchart, Gantt chart, PERT chart, and IDEF have been around since the beginning of the 20th century. Gantt charts were among the first to appear around 1899, flowchart in the 1920s, functional flowblock chart, and PERT in the 1950s, data flowchart, and IDEF in the 1970s. Modern methods include Unified Modeling Language and Business Process Model and Notation, but they are only a fraction of the methods used over the years to document business processes. S. Williams coined the term "business process modeling" in the 1960s in his 1967 article "Business Process Modelling Improves Administrative Control." | Business process modeling techniques such as flowchart, functional flowblock chart, control flowchart, Gantt chart, PERT chart, and IDEF have been around since the beginning of the 20th century. Gantt charts were among the first to appear around 1899, flowchart in the 1920s, functional flowblock chart, and PERT in the 1950s, data flowchart, and IDEF in the 1970s. Modern methods include Unified Modeling Language and Business Process Model and Notation, but they are only a fraction of the methods used over the years to document business processes. S. Williams coined the term "business process modeling" in the 1960s in his 1967 article "Business Process Modelling Improves Administrative Control." |
| **IRP (Ours)** | S. Williams coined the term "business process modeling" in the 1960s in his 1967 article "Business Process Modelling Improves Administrative Control.". These process models help organizations document workflows, surface key metrics, pinpoint potential problems and intelligently automate processes. Business process modeling became the foundation of new methods, for example those that supported data collection, data flow analysis, process flow diagrams and reporting facilities. Business process modeling techniques have been developed over the decades to support specific business needs. | The term "business process modeling" itself was coined in the 1960s in the field of systems engineering by S. Williams in his 1967 article "Business Process Modelling Improves Administrative Control.". His idea was that techniques for obtaining a better understanding of physical control systems could be used in a similar way for business processes. It was not until the 1990s that the term became popular. |
| LED | The term "business process modeling" was coined by the systems engineer S. Williams in 1967. | **Fractal patterns** have been found in the paintings of American artist Jackson Pollock. Pollock's paintings were composed of chaotic dripping and splattering, and Pollock's paintings were composed of chaotic dripping and splattering. |
| RAG | A business process model is a graphical representation of a business process or workflow and its related sub processes. A process model can grow out of procedural maps of day to day operations. | **Business Process Management (BPM)** was originally developed in the 1980s by a group of researchers at the University of Illinois at Urbana-Champaign under the name of Business Process Management. Their goal was to "build a business process model that would allow companies to identify and manage their business processes in a way that would reduce costs and increase productivity. |
| BART | The concept of business process management dates back to the 1960s, with the introduction of the concept of "business process management" in the United States. Business process management was originally developed **in the 1970s by the University of California, Berkeley, and the National Institute of Standards and Technology (NIST).** | The concept of business process management dates back to the 1960s, with the introduction of the concept of "business process management" in the United States. Business process management was originally developed **in the 1970s by the University of California, Berkeley, and the National Institute of Standards and Technology (NIST).** |

Table 12: Baseline comparison of generated expository documents on WikiCS for Business Process Modeling. Red text indicates factual errors. While LED is factually correct, it does not provide enough detail on the *with doc* dataset and fails to discuss the topic of Business Process Modeling adequately on the *without doc* dataset.