# OpenReview forum: "Expository Text Generation: Imitate, Retrieve, Paraphrase"
_EMNLP/2023/Conference — EMNLP 2023 Main_

### Official Review · Reviewer_Q3e3 · 2023-08-01

**Typos Grammar Style And Presentation Improvements:** 83
**Soundness:** 5

**Excitement:**

4: Strong: This paper deepens the understanding of some phenomenon or lowers the barriers to an existing research direction.

**Paper Topic And Main Contributions:**

The paper introduces an architecture to generate stylistically consistent expository texts in a guided manner. The authors introduce a imitate retrieve paraphrase setup to guide the generation. Using a variety of methods (both n-gram overlap based and factualness-based ones) they evaluate their methodology against gold standard documents in  addition to using human evaluation. They test against a wide variety of other approaches, including 7B Llama models for the automated evaluation and ChatGPT for the human evaluation. Ultimatly there model outperforms the comparison in most scenarios.

**Questions For The Authors:**

A) Do I understand it correctly in 3.3 that the training is done using the ground-truth document, i.e. the generated content plan + the facts -> gold sentence? If this is indeed the case, this could be made a bit more explicit. I guess the confusion came because the implicit assumption is that the facts in x need to align with the gold document already. Update: this is explained in 3.5 but should probably be referenced in 3.3 in some way.

B) In the "with doc" scenario are only back-translated versions of the document included, i.e. is the exact original text excluded?

C) Do you think larger models would be in any way beneficial for your approach? (I don't mean to imply that this should have been tested.)

**Reasons To Accept:**

The field of guided generation is a path towards aligning modern LLMs with human expectations and therefore crucially important to the field as a whole.

The experiments seem to be conducted very carefully with a specific focus on not poisoning the retrieval dataset with the ground truth. The evaluation is conducted using a wide range of metrics on top of that even a human evaluation is performed which I would consider unusually thorough. The presented approach performs very well.

The analysis of factual errors was particularly interesting and something I wish I would see in more publications.

**Reasons To Reject:**

Since the dataset is newly introduced the quality of the approach is hard to assess quantitatively, but a good range of comparisons are provided. The IRP approaches applicability to other domains is not explored.

**Reproducibility:**

5: Could easily reproduce the results.

**Reviewer Confidence:**

4: Quite sure. I tried to check the important points carefully. It's unlikely, though conceivable, that I missed something that should affect my ratings.

---

> ### Author Rebuttal · Authors · 2023-08-25
>
> Thank you for the detailed review of our paper! We appreciate that the reviewer finds our work to be crucially important for the field of aligning modern LLMs with human expectations, believes that our experimental setup and evaluation are careful and thorough, and feels that our analysis of factual errors was particularly interesting.
>
> ### RE: Applicability to Other Domains
>
> We appreciate the reviewer’s concern regarding the applicability of our method. For the scope of this work, we focus on the task of expository text generation. We would first like to note that expository text generation alone has several diverse applications other than the ones we studied, such as generating biographies, synthesizing travel guides, or creating product descriptions. Further, we believe that IRP could be adapted for other NLP tasks, such as answering questions in a uniform style in long-form question answering or generating stylistically consistent summaries in multi-document summarization. We feel all of these tasks and applications could become meaningful research directions. We can add a discussion of these applications in the final version.
>
> ### Question A
>
> Yes, we train the paraphraser in section 3.3 to use the generated content plan and retrieved facts as inputs to generate the gold sentence as output. Thank you for noting our explanation in section 3.5. We agree with your suggestion to make section 3.3 clearer and can incorporate more details into section 3.3 in the final version.
>
> ### Question B
>
> In the 'with doc' scenario, we **do not include** the exact gold output as part of the input factual corpus.
>
> ### Question C
>
> We did not explore if larger components would be beneficial for our model. We believe that this may be the case, so in future work, it would be interesting to explore if these incremental quantitative benefits justify the increase in model size.
>
> ### RE: Presentation Improvements
>
> We agree with all of these points! In particular, we will be sure to modify line 83 so that it is more similar to how our content plan (i.e. Imitator) actually functions. Thank you again for taking the time to provide such thorough and insightful comments.

---

### Official Review · Reviewer_yjk6 · 2023-08-04

**Soundness:** 4

**Excitement:**

4: Strong: This paper deepens the understanding of some phenomenon or lowers the barriers to an existing research direction.

**Paper Topic And Main Contributions:**

This paper presents a novel method for generating more factually accurate and stylistically appropriate expository documents. It's called "Imitate, Retrieve, Paraphrase" -- which summarizes the steps:
1. Sketch a template ("content plan") for the next sentence based on stylistic imitation of documents on similar topics
2. Retrieve facts from a knowledge source to fill in factual statistics/statements
3. Paraphrase the retrieved facts to match the style of the content plan.

The main contributions are:
- Proposes the IRP framework as a general method for generating expository documents
- Publishes three datasets to facilitate further research on factuality and style in text generation
- Shows via experiments that IRP produces more factual and stylistically appropriate expository documents than competing models

**Reasons To Accept:**

While the IRP method uses pretrained models (e.g. DistilBERT, GPT-2, etc), it composes them in a very interesting way in order to produce a framework for expository document generation.

Along the way, some very smart choices are made. For example, the authors propose an interesting method for generating content plans, by optimizing DistilBERT to predict the index of a sentence in a document (which thereby lowers token attribution for facts which vary widely between documents).

Expository text generation is a practical task, and the methods proposed here for structuring content plans and retrieving facts can be applied to many other types of text generation as well.

**Reasons To Reject:**

I can't see any. it's a really nice paper.

**Reproducibility:**

4: Could mostly reproduce the results, but there may be some variation because of sample variance or minor variations in their interpretation of the protocol or method.

**Reviewer Confidence:**

4: Quite sure. I tried to check the important points carefully. It's unlikely, though conceivable, that I missed something that should affect my ratings.

---

> ### Author Rebuttal · Authors · 2023-08-25
>
> Thank you for your review and we appreciate your support of our work! We are delighted to hear that the reviewer found our design choices and composition of pretrained models to be very interesting and smart. Further, it's encouraging to hear that our techniques for generating content plans and retrieving facts were perceived to have applicability in other text generation tasks, which we also feel would be very interesting future work. We are grateful for the time and effort you've taken to thoroughly review our paper and provide insightful feedback.

---

### Official Review · Reviewer_7sKR · 2023-08-07

**Soundness:** 4

**Excitement:**

4: Strong: This paper deepens the understanding of some phenomenon or lowers the barriers to an existing research direction.

**Paper Topic And Main Contributions:**

The paper is about an iterative three-step technique for expository text generation. It introduces the task and offers a framework and datasets. It also presents a thorough experimental comparison on automatic metrics and has a human evaluation component.

**Reasons To Accept:**

The evaluation seems thorough, and the technique proposed seems to have advantages for both interpretability and factuality. The datasets could be useful for future research.

**Reasons To Reject:**

The three-step nature of the proposed architecture could result in error propagation.

**Reproducibility:**

5: Could easily reproduce the results.

**Reviewer Confidence:**

3: Pretty sure, but there's a chance I missed something. Although I have a good feel for this area in general, I did not carefully check the paper's details, e.g., the math, experimental design, or novelty.

---

> ### Author Rebuttal · Authors · 2023-08-25
>
> Thank you for your review! We appreciate that the reviewer finds our experimental evaluation to be thorough, notes that our method has advantages in interpretability and factuality, and believes our datasets could be useful for future research.
>
> ### RE: Error Propagation
>
> We appreciate the reviewer’s concern regarding error propagation in our model. We agree that the three-step nature of our approach could leave some room for error propagation. However, we investigated this weakness of error propagation in Section 5.4 and did not find it significant. Specifically, we “find that many factual errors are due to alternatives existing in the retrieved facts, rather than the weaknesses of the Retriever, Paraphraser or data collection” (lines 573-576).
>
> We also acknowledge this shortcoming in the limitation section. We decided not to train an end-to-end model because it made our approach more interpretable and thus easier to detect errors in each individual component (lines 613-616). Further, we believe the fact that IRP uses three separate components instead of an end-to-end framework suggests that there is still room for improvement and research in expository text generation (lines 611-619).

---

### Meta-Review · Area_Chair_fkQi · 2023-09-19

**Recommendation:** 5

**Metareview:**

This paper proposes the task of expository text generation and its evaluation dataset. Furthermore, the authors also propose the method named "Imitate, Retrieve, Paraphrase" (IRL) for generating more factually accurate and stylistically appropriate expository documents. This three-step approach has advantages for both interpretability and factuality in generating text. The experimental results on the newly created dataset by the authors show that IRP produces more factual and stylistically appropriate expository documents than competing models. The authors answered all questions by the reviewers, and all reviewers agreed with the acceptance of this paper. Thus, we should follow their agreement.

---

### Decision · Program_Chairs · 2023-10-07

**Decision:**

Accept-Main

**Comment:**

This paper proposes the task of expository text generation and its evaluation dataset. Furthermore, the authors also propose the method named "Imitate, Retrieve, Paraphrase" (IRL) for generating more factually accurate and stylistically appropriate expository documents. This three-step approach has advantages for both interpretability and factuality in generating text. The experimental results on the newly created dataset by the authors show that IRP produces more factual and stylistically appropriate expository documents than competing models. The authors answered all questions by the reviewers, and all reviewers agreed with the acceptance of this paper. Thus, we should follow their agreement.